# CONCENTRIC SPHERICAL GNN FOR 3D REPRESENTATION LEARNING

## ABSTRACT

Learning 3D representations that generalize well to arbitrarily oriented inputs is a challenge of practical importance in applications varying from computer vision to physics and chemistry. We propose a novel multi-resolution convolutional architecture for learning over concentric spherical feature maps, of which the single sphere representation is a special case. Our hierarchical architecture is based on alternatively learning to incorporate both intra-sphere and inter-sphere information. We show the applicability of our method for two different types of 3D inputs, mesh objects, which can be regularly sampled, and point clouds, which are irregularly distributed. We also propose an efficient mapping of point clouds to concentric spherical images using radial basis functions, thereby bridging spherical convolutions on grids with general point clouds. We demonstrate the effectiveness of our approach in achieving state-of-the-art performance on 3D classification tasks with rotated data.

## 1 INTRODUCTION

While convolutional neural networks have been applied to great success to 2D images, extending the same success to geometries in 3D has proven more challenging. A desirable property and challenge in this setting is to learn descriptive representations that are also equivariant to any 3D rotation. Cohen et al. (2018) and Esteves et al. (2018) showed that the spherical domain permits learning such rotationally equivariant representations, by defining convolutions with respect to spherical harmonics. In practice, 3D convolutions are implemented via discretization of the sphere. Earlier spherical Convolutional Neural Networks (CNNs) used spherical coordinate grids, but these discretizations result in non-uniform samplings of the sphere, which is non-ideal. Furthermore, spherical convolutions defined on these grids scale with $O(N^{1.5})$ complexity ($N$ as the number of grid points). Subsequent works, Jiang et al. (2019), Cohen et al. (2019), Defferrard et al. (2020), designed more scalable $O(N)$ convolutions focusing on more uniform spherical discretizations.

Existing spherical CNNs operate over a spherical image, resulting from projection of data to a bounding sphere. We show that it is more expressive and general to instead operate over a concentric, multi-spherical discretization for representing 3D data. Our main innovation is introducing a new two-phase convolutional scheme for learning over a concentric spheres representation, by alternating between inter-sphere and intra-sphere convolutional blocks. We use graph convolutions to incorporate inter-sphere information, and 1D convolutions to incorporate radial information. Similar to Jiang et al. (2019) and Cohen et al. (2019), we focus on the icosahedral spherical discretization, which produces a mostly regular sampling over the sphere. Our proposed architecture is hierarchical, following the recursive coarsening hierarchy of the icosahedron. Combining intra-sphere and inter-sphere convolutions has a conceptual analogy to gradually incorporating information over volumetric sectors. At the same time, the choice of convolutions allows our model to retain a high degree of rotational equivariance.

We demonstrate the effectiveness and generality of our approach through two 3D classification experiments with different types of input data: mesh objects and general point clouds. The latter poses an additional challenge for discretization-based methods, as native point clouds are non-uniformly distributed in 3D space.

To summarize our contributions:

1. We propose a new multi-sphere icosahedral discretization for representation of 3D data, and show that incorporating the radial dimension can greatly enhance representation ability over single-sphere representations.
2. We also introduce a novel convolutional architecture for multi-sphere discretization by introducing two different types of convolutions, conceptually separated as intra-sphere and inter-sphere. Combining graph convolutions (intra-sphere) with 1D radial convolutions (inter-sphere) leads to an expressive architecture that is also rotationally equivariant. Our proposed convolutions are also scalable, being linear with respect to total grid size.
3. We design mappings of both 3D mesh objects and general point clouds to the proposed representation. We achieve state-of-art performance on ModelNet40 point cloud classification, using the proposed model and a data mapping using radial basis functions. We also improve on existing Spherical CNN performance in SHREC17 3D mesh classification by utilizing multi-radius information.

## 2 RELATED WORK

**Spherical CNNs.** The goal of learning rotationally invariant representations of 3D geometries has led to several ideas for rotationally equivariant convolutions in the spherical domain. Cohen et al. (2018) and Esteves et al. (2018) defined spherical convolutions that are rotationally equivariant to rotations of the $SO(3)$ group. However, these convolutions are restricted to non-uniform grid samplings and scale superlinearly with respect to grid resolution. Later works have explored more scalable convolutions on other spherical discretizations, achieving linear complexity with respect to grid resolution.

Jiang et al. (2019) proposed using parameterized differential operators to form convolutional kernels over the icosahedron, where equivariance is restricted to rotations about the $z$-axis. Cohen et al. (2019) proposed gauge equivariant convolutions on manifolds, operating on feature fields corresponding to underlying geometric entities. This was applied to achieve rotationally equivariant convolutions over the icosahedral discretization. Defferrard et al. (2020) propose a graph convolution-based spherical CNN using spectral filters, along with a distance-weighted nearest-neighbors graph construction scheme that allows balancing between rotational equivariance and efficiency, when applied to different types of grids.

Other spherical CNNs have been designed in the context of handling arbitrary point cloud data, which typically requires first mapping the data to a discretization. Rao et al. (2019) uses graph-convolution inspired message passing operators for learning over the icosahedral discretization. Our work is similar to Rao et al. (2019) and Defferrard et al. (2020)) in terms of using graph-based spherical convolutions, but we generalize to multi-sphere convolutions. You et al. (2020) is the most related work in terms of multi-sphere representation learning. The authors propose a spherical voxel grid, and extending the SO3 convolutions of Cohen et al. (2018) to incorporate the radial dimension. Our work treats spherical and radial convolutions as distinct, which results in much better results in practice. We also use more scalable spherical convolutions defined on the uniform icosahedral grid.

**Pointwise Convolution Networks.** There is a significant body of work on learning point cloud representations using pointwise convolutions, beginning with with Qi et al. (2017) which proposed learning permutation invariant functions that directly operate on point coordinates. Only more recently have such methods have been developed towards learning rotationally invariant representations. Thomas et al. (2018) and Poulenard et al. (2019) both propose pointwise convolutional filters based on spherical harmonic functions to achieve rotational equivariance (or invariance). Distance information is recorded through learned functions in the former, and radial sampling in the latter. While these filters are defined with respect to all-to-all convolution between points, in practice convolutions are limited to $k$-nearest neighbors (Poulenard et al. (2019)) for scalability. Chen et al. (2019), Sun et al. (2019), Zhang et al. (2019) all extract rotationally invariant features (i.e. low-level geometric features such as angles and distances) from the point cloud as input to their respective convolutional architectures. These features are hand-engineered based on carefully picking local frames of references, or global ones in the case of Sun et al. (2019).

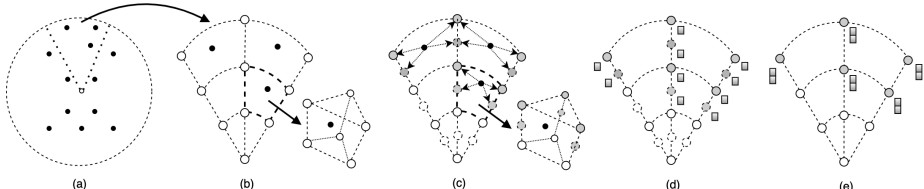

Figure 1: (a) shows an example point cloud (black dots) contained within a bounding sphere. (b) shows the spherical partioning of 3D space in 2D cross section view, and zooms in on a sector occupied by 3 data points. Vertices are white circles. Each point is bounded by a neighborhood of 6 vertices, 3 from the sphere above and 3 from below. (c),(d) Each point is mapped to scalar values defined on the bounding vertex neighborhood using radial basis functions. Vertices affected by the mapping are shaded gray. Dotted circles indicate vertices temporarily added in the radial dimension to increase resolution. (e) Vertex values are concatenated into feature channels of original vertices.

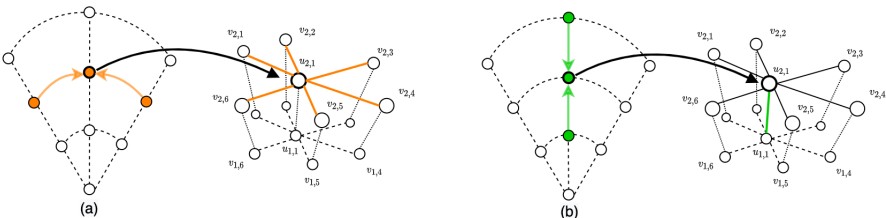

Figure 2: Two subsets of vertices from two concentric spheres, connected radially. $u_{i,1}$ or $v_{i,*}$ are vertices on i-th sphere (a) Intra-sphere convolution and (b) inter-sphere convolution applied with respect to the target vertex $u_{2,1}$ (bolded). Third sphere not shown for clarity. Vertices involved in convolution are connected by orange or green edges.

## 3 REPRESENTATION BY CONCENTRIC SPHERES

Existing work on spherical CNNs operate on spherical grids, where data is typically projected to and defined on grid points. However, projecting 3D data to a single sphere may not always be sufficient or appropriate. Simple projections may be lossy when describing highly non-convex shapes, for instance if the shape curves in on itself. To increase capacity to distinguish different data distributions, we introduce a new discretization based on concentric spheres, which additionally discretizes 3D space in the radial dimension. The single sphere discretization is a special case in our proposed paradigm.

**Spherical Discretization.** We work with an icosahedral grid discretization of the sphere. The base icosahedron $I^{(0)}$ has 12 vertices, forming 20 equilateral triangle faces (each face with 3 edges). Each vertex is incident with 5 triangles. Each face can be subdivided, with the number of vertices scaling as $|V| = 10 * 4^l + 2$ (where $l$ is the discretization level). See Fig. 3 for an illustration.

**Radial Discretization.** We construct the multi-radius spherical discretization by stacking $R$ identical icosahedral grids. Assuming unit radius normalization, we use a uniform discretization that results in concentric spheres scaled to radii $[\frac{1}{R}, \frac{2}{R}, ..., 1]$.

**Intra-sphere Convolutions.** There is a growing body of work addressing design of rotation-equivariant filters over spherical feature maps. We focus on graph convolutional filters for intra-sphere convolutions, as graph convolutions are scalable and lead to equivariant representations, up to discretization effects Defferrard et al. (2020), Yang et al. (2020). This motivates our construction of the undirected graph $G^{(l)} = (V^{(l)}, E^{(l)})$ from a level $l$ icosahedron $I^{(l)}$. Vertices of the vertex set $V^{(l)}$ correspond one-to-one with vertices of $I^{(l)}$ projected to unit sphere. $E^{(l)}$ is simply the set of all (bidirectional) face edges of the icosahedron (projected to unit sphere). Vertices are all degree 6, with exception of the the initial 12 vertices of the base icosahedron $I^{(0)}$ that are degree 5. Since each edge is approximately equidistant between two points of the sphere Wang & Lee (2011), $G^{(l)}$ is also treated as an unweighted graph.

While there is a rich body of work on graph convolutions and its variants, this work focuses on the graph convolution defined in Kipf & Welling (2017). We introduce notation to define this convolution in the context of our multi-spherical discretization in more detail. Let $\mathbf{H} \in \mathbb{R}^{R \times |V| \times C}$ denote a $C$ channel tensor of features. Also let $\boldsymbol{Z} \in \mathbb{R}^{C \times F}$ be shared parameters, where $C$ and $F$ are input and output channel sizes. $N(u)$ denote neighbors of vertex $u$ in graph $G$ and $\deg(u)$ denotes $|N(u)|$, the degree of vertex $u$. We assume that self-edges are added for every vertex in $N(u)$. Finally, we introduce the subscript $t$ to be the convolutional layer number, $i \in [0, R-1]$ to index the radial dimension, and $u \in [0, |V|-1]$ to index the vertices. The layer $t+1$ intra-sphere convolution output for vertex $u$ of sphere $i$ is given by Eq. 1, where $\sigma$ indicates a nonlinear activation function.:

$$\mathbf{H}_{i,u}^{(t+1)} = \sigma\left( \sum_{v \in N(u)} \frac{1}{\sqrt{\deg(u)\deg(v)}} \mathbf{H}_{i,v}^{(t)} \boldsymbol{Z}^{(t)} \right) \tag{1}$$

**Inter-sphere Convolutions.** We introduce *radial convolutions* to incorporate inter-sphere information, implemented as 1D convolutions where the radial dimension is treated as the sequence length. Importantly, radial convolutions are also rotationally invariant, as 1D convolution operates over channels of the same vertex. See Fig. 2 for illustration of graph convolutions with respect to concentric spheres representation. We introduce some additional notation to describe radial convolutions. Let $K$ be 1D convolution kernel size. We assume $K$ is odd valued, and pad inputs in the radial dimension such that a dimension of $R$ is maintained across convolutions. Let $\boldsymbol{W} \in \mathbb{R}^{K \times C \times F}$ be a tensor of shared parameters, where $C$ and $F$ are input and output channel sizes. The layer $t+1$ inter-sphere convolution output for vertex $u$ of sphere $i$ is given by Eq. 2:

$$\mathbf{H}_{i,u}^{(t+1)} = \sigma\left( \sum_{k=-\lfloor\frac{K}{2}\rfloor}^{\lfloor\frac{K}{2}\rfloor} \mathbf{H}_{i+k,u}^{(t)} \mathbf{W}_{k+\lfloor\frac{K}{2}\rfloor}^{(t)} \right) \tag{2}$$

**Vertex Pooling.** Pooling is widely used alongside convolutional filters in CNN architectures to learn invariance to transformations of the input. The icosahedron, due to its recursive refinement by discretization level, defines a natural hierarchy for pooling and downsampling (see Fig. 3). We introduce overloaded notation $\mathbf{H}^{(l)} \in \mathbb{R}^{R \times |V^{(l)}| \times C}$ to denote the feature tensor corresponding to $V^{(l)}$, the vertex set corresponding to level $l$ icosahedron. We define pooling as $\mathbf{H}_{i,u}^{(l-1)} = f(\{\mathbf{H}_{i,v}^{(l)} : v \in N(u)\})$, where $N(u)$ is the neighborhood of vertex $u \in V^{(l)}$ and $f$ is a permutation invariant function (e.g. max operator). Pooling is followed by downsampling, where only vertices of the smaller vertex set $V^{(l-1)}$ are retained (see Sec. A.4 for details.) Only vertices within same sphere are involved; there is no pooling or downsampling of vertices between spheres (except right before classification).

**Concentric Spherical GNN (CSGNN) Architecture.** Fig. 3 gives an example illustration of an end-to-end architecture using both convolutions. Importantly, radial convolution blocks are introduced alongside graph convolutional blocks at every level of the spherical discretization hierarchy, to incorporate inter-sphere information gradually. From a icosahedron of level $L$ refinement, we construct a sequences of graphs $[G^{(L)}, G^{(L-1)}, ..., G^{(0)}]$. Each $G^{(l)}$ carries an additional $R$ dimension, corresponding to spheres at different radial levels. Each level $l$ features two blocks of convolutions: graph convolutions, followed by radial convolution. These correspond to intra-sphere and inter-sphere convolutions respectively. Vertex neighborhood pooling downsamples the graph from $G^{(l)}$ to $G^{(l-1)}$. The size of the radial dimension remains constant, until final pooling.

**Complexity Analysis.** In the intra-sphere convolution of Eq. 1, the neighborhood size (vertex degree) is bounded by a constant. Dimensions of parameters and feature vectors are also bounded by a constant. Therefore, the overall complexity of intra-sphere convolution is linear with respect to the multi-spherical grid size, or $O(R|V|)$, where $|V|$ is the number of vertices in the spherical grid. $R$ is can be effectively kept very small relative to $|V|$, as we show in our experiments. Similarly, each inter-sphere convolution (Eq. 2) with respect to a vertex is bounded a fixed kernel size for the neighborhood. The overall complexity of inter-sphere convolution is also $O(R|V|)$, and so both convolutions introduced in this work are linear with respect to the multi-spherical grid size. We further provide experimental parameter analysis in Tables 1, 2, and time analysis in Appendix A.2.

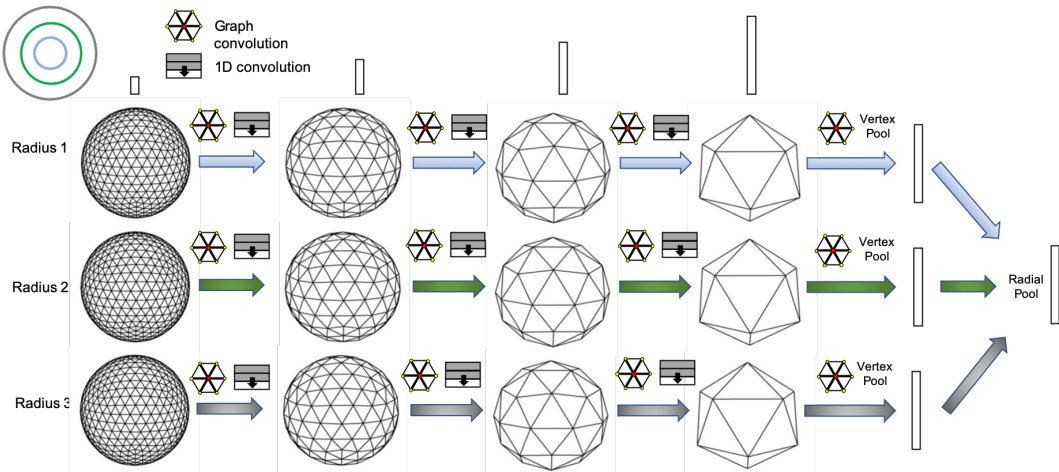

Figure 3: Example multi-radius architecture with $R = 3$ concentric spheres. Graph convolutions, followed by radial convolutions, are applied over a sequence of discretization levels. Pooling coarsens the discretization to a lower level. Vertex-wise and radial-wise pooling is applied to obtain a final representation for classifier. Icosahedron visualization from Satoh et al. (2014).

## 4    POINT CLOUD TO CONCENTRIC SPHERICAL SIGNAL

We consider the problem of mapping a point cloud $\boldsymbol{P} \in \mathbb{R}^{N \times 3}$ point cloud to an initial spherical feature map $\mathbf{M} \in \mathbb{R}^{R \times |V| \times C}$, where $C$ is number of input channels. While the concentric grid representation is defined discretely at fixed positions, the space of data point locations is continuous. We also aim to capture the distribution of points in a continuous way. To do so, we summarize the contribution of points using the Gaussian radial basis function (RBF):

$$f(\boldsymbol{x}) = \sum_{j=1}^{N} \phi(||\boldsymbol{x} - \boldsymbol{P}_j||_2^2) \tag{3}$$

$N$ is the number of data points, and $\phi = \exp(-\gamma r^2)$, parameterized by the bandwidth $\gamma$. In practice we limit computation to a local neighborhood (instead of considering all points), and choose $\gamma$ accordingly. See Fig. 1 for visualization of the local neighborhood and mapping, and Sec. A.1 for additional details.

One possible mapping is to compute Eq. 3 at every vertex position of the spherical discretization, resulting in a single channel feature map. However, it is possible to obtain better resolution in capturing distribution of surrounding points by further sub-diving the discretized space, taking inspiration from Meng et al. (2019), along the radial dimension. Subdividing along radial dimension by a factor $K_e$ results in a new spherical discretization with increased radial dimension of $R' = R * K_e$. The RBF is evaluated at every vertex position of this new discretization, resulting in a feature map of dimension of $\mathbf{M}' \in \mathbb{R}^{R' \times V \times 1}$. We map back to the original discretization by assigning $\mathbf{M}_{i,u} = [\mathbf{M}'_{j,u} : j \in (iK_e, iK_e + 1, ..., 2iK_e, 2iK_e + 1, ..., 3iK_e - 1)]$, resulting in a size $[R \times |V| \times 2K_e]$ spherical feature map. In summary, multiple RBF values are assigned to each vertex by further sub-dividing space in the radial dimension.

## 5    EXPERIMENTS

### 5.1    MODELNET40 POINT CLOUD CLASSIFICATION

We consider the ModelNet40 3D shape classification task, with 12308 shapes and 40 classes. Each point cloud has 1024 points. For all experiments, 9840 shapes are used for training and 2468 for testing.

**Architecture and Hyperparameters**
Fig. 4 shows a complete architecture overview. Point clouds are first mapped to 16 concentric

spheres with level 4 icosahedral discretization ($L = 4, R = 16$), using RBF kernels with threshold $T = 0.01$. 1D convolutions use a kernel size of 3. Each graph and 1D convolution is followed by batch normalization and ReLU as nonlinear activation. Additionally, skip connections are added between every graph convolution layer, whenever input dimension matches output. The model is trained with Adam optimizer for 30 epochs using initial learning rate of 1e-3, along with learning rate decay by 0.1 at 15 and 25 epochs. Batch size is 32.

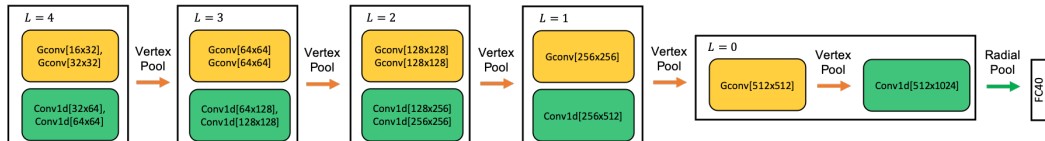

Figure 4: Architecture for ModelNet40 classification. Input dimension is 16, resulting from point cloud RBF data mapping. "Gconv" is graph convolution applied over graph connectivity of the sphere. "Conv1d" is 1D convolution, applied over the radial dimension. $L$ denotes discretization level, as the representation is coarsened following each vertex pooling step. A final pooling of radial dimension results in a 1024 dimensional vector.

**Results**

| Method | Input | Params | z/z | z/SO3 | NR/SO3 | SO3/SO3 |
|---|---|---|---|---|---|---|
| *Pointwise Convolution* | | | | | | |
| PointNet Qi et al. (2017) | xyz | 3.5M | 0.875 | 0.229 | 0.081 | 0.849 |
| ClusterNet Chen et al. (2019) | xyz | * | 0.871[1] | 0.871[1] | * | 0.871[1] |
| RIConv Zhang et al. (2019) | xyz | 0.7M | 0.870 | 0.870 | **0.872** | 0.872 |
| SPHNet Poulenard et al. (2019) | xyz | 2.9M | 0.865 | 0.856 | 0.854 | 0.870 |
| SRINet Sun et al. (2019) | xyz+ normal | 0.9M | 0.844 | 0.829 | 0.834 | 0.837 |
| *Spherical CNN* | | | | | | |
| SFCNN Rao et al. (2019) | xyz | 9.2M | **0.888** | 0.831 | 0.350 | 0.874 |
| PRIN You et al. (2020) | xyz | 1.7M | 0.819 | 0.765 | 0.753 | 0.810 |
| **Ours** (CSGNN) | xyz | 2.8M | 0.884 | **0.874** | 0.833 | **0.884** |

Table 1: ModelNet40 classification results, across four train/test data orientation settings. NR denotes original data (no rotations), $z$ is arbitrary rotation about $z$ axis, and $SO3$ is arbitrary rotation. For example, SO3/SO3 means training and testing with arbitrary rotations of the data. Params is number of parameters in millions.

We present our results and compare against other related works in Table 1, in four different train/test data orientation settings. When training with rotations, a new rotation is sampled per instance in each epoch. Rotation is the only augmentation used in comparisons. We report accuracy as average validation score across last 5 epochs of training, due to lack of standard validation/test split and to account for variation.

Our method achieves state-of-the-art results in z/SO3 and SO3/SO3 settings, i.e. testing on arbitrarily rotated data. For more detailed comparison, we loosely categorize compared works by method into two categories: pointwise convolution networks and spherical CNNs. Methods in the former category operate directly on data points in 3D space, while methods in the latter operate on a spherical discretization. Our work is most closely related to methods in the spherical CNN category.

Similar to our work, PRIN also explored learning a concentric spherical representation based on extending SO3 convolutions from Cohen et al. (2018). Our method is based on separate graph and radial convolutions, which achieves much better performance in all settings. SFCNN has similarity to our work in using graph convolution-inspired message passing filters and learning over the icosahedral discretization of the sphere. However, SFCNN is restricted to a single-sphere representation, and also relies on a PointNet-like learned module to project points to spherical features.

---

[1]Using available numbers reported in authors' paper, as the authors' code was not publically available at the time of this writing.

While this learned projection should be able capture some degree of multi-radius information in the point distribution, best results seem to be achieved by learning from both intra-sphere and inter-sphere convolutions. Compared to our approach, SFCNN also has a relatively higher performance gap between z/z and any SO3 test setting (significantly so for z/SO3 and NR/SO3), which suggests greater difficulty achieving rotational invariance. CSGNN, similar to SFCNN and PRIN, exhibits some drop in performance in the z/SO3 and NR/SO3 settings. This could partly be due to effects of discretization and data mapping. However, this gap is relatively smallest in CSGNN, and there is essentially no gap in z/z and SO3/SO3 performance.

ClusterNet, RIConv, and SRINet use hand-crafted rotationally invariant geometric features as inputs, and so there is negligble to no performance gap in testing with or without rotations. By contrast, our method largely learns to extract features directly from the input (outside of an initial step mapping points to vertices). PointNet, unlike the other baselines, was not designed to be rotationally equivariant. This reflects in the relatively large difference in performance when comparing testing with or without rotations. Even when training with rotations and using a learned alignment module that attempts to learn a canonical transformation, PointNet SO3/SO3 performance is not competitive with that of most other baselines.

## 5.2 SHREC17 3D Shapes

The SCHREC17 task has 51300 3D models and 55 categories. We use the version where all models have been randomly perturbed by rotations. Here the inputs are not point clouds, but mesh objects. Cohen et al. (2018),Esteves et al. (2018) presented a ray-casting scheme to regularly sample information incident to outermost mesh surfaces and obtain features maps defined over the spherical discretization. For sufficiently non-convex mesh objects, a single sphere projection may result in information loss. For example, when a ray is incident to multiple surfaces occurring at different radii, this information is discarded by existing methods. We propose a new data mapping that generalizes single sphere representation to a concentric spherical representation, thereby preserving more information. Fig. 8 in the appendix shows visual examples of where the proposed representation may be helping.

**Representation**

In the case of single-sphere representations, a single ray is projected from a source point (vertex) on the enclosing sphere towards the center of the object. The first hit incident with the mesh is recorded. To extend ray-casting to multiple concentric spheres, we rescale the source point to the radii of each respective sphere. This results in multiple co-linear source points, one per sphere. The 1st hit incident with the mesh is recorded for each ray cast from those source points, resulting in a multi-radius projection. While this new scheme is not sufficient to capture all incident surface information (e.g. if there are multiple hits sandwiched between two radial levels), it provides more samples that scales with the number of spheres. We use a uniform $[\frac{1}{R}, \frac{2}{R}, ..., 1]$ radii division assuming inputs are normalized to unit radius. From each point of intersection with the mesh, the distance (with respect to outermost sphere) to the point of incidence as well as $sin$ and $cos$ features are recorded, resulting in 3 features per vertex. These are similar features to those collected in related work, except we do not use the object's convex hull information.

| Method | Params | $F_1$ |
|---|---|---|
| Cohen et al. (2018) (equiangular, $b = 64$) | 0.4M[2] | 0.789[2] |
| Esteves et al. (2018) (equiangular, $b = 64$) | 0.5M[2] | 0.794[2] |
| DeepSphere Defferrard et al. (2020) (equiangular, $b = 64$) | 0.2M[2] | 0.794[2] |
| DeepSphere Defferrard et al. (2020) (HEALPix, $N_{side} = 32$) | 0.2M[2] | 0.807[2] |
| CSGNN (icosahedral, $L = 4, R = 1$) | 1.3M | 0.805 |
| CSGNN (icosahedral, $L = 4, R = 16$) | 2.9M | 0.823 |

Table 2: SHREC17 classification performance in terms of $F_1$ metric (micro-average). CSGNN is our implementation. Equiangular, HEALPix, and icosahedral are different discretizations of the sphere. CSGNN (this work) uses level 4 icosahedral discretization, $R$ is number of concentric spheres (specific to this work). Params is number of parameters in millions.

---

[2]Numbers as reported in Defferrard et al. (2020).

**Architecture and Hyperparameters**

The architecture for SHREC17 is identical to the one used for ModelNet40 in Fig. 4, with the exception that the input dimension is 3 (corresponding to features obtained from ray-casting). We consider two model variations, single-sphere ($R = 1$) and multi-sphere ($R = 16$). For $R = 16$, we use a 1D convolution kernel of size 3. For $R = 1$, we use a 1D convolution kernel of size 1. Note this is equivalent to applying fully connected layers; we found adding additional fully connected layers after graph convolutions helped improve performance in the single-sphere case. Training settings are also same as for ModelNet40, except learning rate decays at epochs 10 and 20.

**Results**

See Table 2 for classification results. The reported metric is $F_1$ micro-average classification score. Results from three other Spherical CNN works designed for general rotational equivariance are included for reference. DeepSphere is most similiar to our work in terms of graph-based spherical convolutions, where the authors explored design of rotationally-equivariant graph convolutional filters with respect to the type of grid and neighborhood size. This work focuses graph construction to the icosahedral discretization, using a minimal set of roughly equidistant neighbors.

We additionally introduce inter-sphere convolutions with concentric spheres, which is largely orthogonal to the design of intra-sphere representation and convolutions. Single-sphere ($R = 1$) CSGNN achieves competitive performance with other single-sphere baselines. However, it is difficult to draw comparative conclusions in this particular case, due to differences in feature extraction, spherical discretization type and size, and model size. More significantly, using multiple spheres ($R = 16$) outperforms single-sphere baselines, including a 2.2% relative performance improvement over our $R = 1$ version. It also seems likely that the concentric spheres approach can improve other spherical convolutional designs as well, but this is beyond the scope of this work.

## 5.3 MODEL ARCHITECTURE ABLATION STUDY

To study the impact of multi-radius spherical discretization, we vary the number of radial levels and present results in Table 3. ModelNet40 is used for all ablation experiments. We use a base model with $R = 16$ and $L = 4$, and keep the number of parameters identical across all versions of the model. For this particular ablation, for simplicity we use a single channel, indicator feature map–a special case of the RBF mapping where $\gamma = 0$ and $F = 1$. Adding radial convolutions in the case of $R = 1$ is equivalent to adding additional dense layers after graph convolutions. The architecture is the same as the one in 4, except input dimension is 1. Performance consistently improves with higher radial dimension, peaking at $R = 16$ with 4.8% relative accuracy improvement over the $R = 1$ version. Performance declines from $R = 16$ to $R = 32$, which suggests diminishing returns for a fixed parameter budget.

| Setting | $R = 1$ | $R = 4$ | $R = 8$ | $R = 16$ | $R = 32$ |
|---------|---------|---------|---------|----------|----------|
| SO3/SO3 | 0.839 | 0.857 | 0.869 | 0.879 | 0.872 |

Table 3: ModelNet40 ablation with number of radial levels ($R$). Number of model parameters is fixed across all settings.

More ablation studies are presented in Table 4. We study the impact of varying the radial kernel size $K_{RC} = [1, 3, 5]$. $K_{RC} = 1$ is same as learning representations independently learned at each radial level. While this still improves over single-sphere representation, using spatial filters ($K_{RC} = [3, 5]$) over the radial dimension is important for best performance. Varying the number of graph and radial convolutional layers shows that between 1 and 2 layers per block leads to comparable performance. Finally, we compare using only graph convolutions or only radial convolutions. Results suggest that it is essential to combine both types of convolutions for best performance. Interestingly, restricting to radial convolutions achieves slightly better performance than restricting to graph convolutions over the single sphere. This provides further empirical support for the expressiveness of our proposed representation and radial convolutions.

## 5.4 POINT CLOUD DATA MAPPING ABLATION STUDY

In this section we compare alternative choices of neighborhood in the mapping of points to vertices, as well as alternative functions to the Gaussian radial basis function (RBF). The bounding vertices

| Setting | SO3/SO3 |
|---|---|
| *Radial kernel size* | |
| $K_{RC} = 1, M_{GC} = 1, M_{RC} = 1$ | 0.853 |
| $K_{RC} = 3, M_{GC} = 1, M_{RC} = 1$ | 0.880 |
| $K_{RC} = 5, M_{GC} = 1, M_{RC} = 1$ | 0.882 |
| *Graph convolution only* | |
| $R = 1, M_{GC} = 1$ | 0.837 |

| Setting | SO3/SO3 |
|---|---|
| *Convolution layers* | |
| $K_{RC} = 3, M_{GC} = 1, M_{RC} = 1$ | 0.880 |
| $K_{RC} = 3, M_{GC} = 2, M_{RC} = 2$ | 0.876 |
| *Radial convolution only* | |
| $K_{RC} = 3, M_{RC} = 1$ | 0.845 |

Table 4: Ablation study on ModelNet40. $K_{RC}$ is size of radial convolutional kernel, $M_{GC}$ and $M_{RC}$ are number of graph and radial convolutional layers per block. $R = 16$ and $L = 4$, unless stated otherwise.

| Name | Function | SO3/SO3 |
|---|---|---|
| *Constant* | $\phi(r) = 1$ | 0.879 |
| *Linear* | $\phi(r) = max(1 - \frac{r}{c}, 0)$ | 0.877 |
| *Inverse Quadratic* | $\phi(r; \gamma) = \frac{1}{1+\gamma r^2}$ | 0.882 |
| *Gaussian* | $\phi(r; \gamma) = exp(-\gamma r^2)$ | 0.884 |

| Neighborhood | SO3/SO3 |
|---|---|
| k-NN, $k = 4$ | 0.875 |
| k-NN, $k = 8$ | 0.870 |
| k-NN, $k = 16$ | 0.875 |
| Bounding vertices | 0.884 |

Table 5: Ablation study on mapping point clouds to concentric spherical signal with ModelNet40. Left table compares different decay functions. Right table compares k-nearest neighbor approach with bounding vertices approach. The former is vertex-centric, considering points are neighbors, whereas the latter is point-centric, influencing its neighboring vertices.

approach (detailed in Sec. A.1) is point-centered, in that each data point influences its immediate surrounding vertices. These vertices form two triangles which "bound" the point in vertices in space (see Fig. 1) corresonding to a neighborhood of 6 vertices. An alternative is to define vertex-centered neighborhood consisting of the $k$-nearest points to each vertex. We use the Gaussian RBF to map the contribution of each point based on its distance to the vertex. The Gaussian parameter $\gamma$ is chosen such that the contribution of points up to a maximum distance of $\frac{2}{R}$ decay to a small threshold value ($T = 0.01$ in this case). Results from Table 5 show the bounding vertices approach outperforming the vertex-centered k-NN approach. One possible reason is the difficulty in controlling for scale, especially when combined with a fixed-parameter decay function, as vertices closer to the center are more densely situated in space compared to vertices further from the center.

We also compare four different decay functions applied to bounding vertices neighborhood. The constant function is trivial in that it is independent to distance, and essentially serves as an indicator. The linear function decays to zero with distance, but is non-smooth. $c$ is a specified parameter for a distance cutoff. Inverse quadratic and Gaussian are two smoothly decaying functions commonly used as radial basis functions. The latter two perform slightly better than the constant and linear functions, suggesting some advantage to using RBFs to summarize neighborhood contributions.

## 6 DISCUSSION AND CONCLUSIONS

In this work we proposed a new multi-sphere convolutional architecture, CSGNN, for learning rotationally invariant representations of 3D data. We introduced distinct intra-sphere and inter-sphere convolutions, which can be combined to learn more expressive representations compared to being restricted to single-sphere representation. Our use of graph and 1D convolutions preserves rotational equivariance, while achieving linear scalability with respect to size of discretization. We achieve state-of-the-art performance in ModelNet classification for testing on arbitrary rotations among both spherical CNN and pointwise convolutional models. We also show that our approach generalizes to classification of 3D mesh objects by improving on single-sphere representation and performance for the SHREC17 task. One avenue of future work is to explore more descriptive mappings of point cloud data to the discretization. A learned assignment may better learn vertex features for describing nearby points. There is also room to explore other kinds of convolutions for incorporating inter-sphere information, as well as other radial division schemes. Finally, existing implementations in this work can be more efficient implementations based on using regular properties of the icosahedral grid, as opposed to using a general graph construction.

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

# A  APPENDIX

## A.1  POINT CLOUD TO SPHERICAL SIGNAL

Instead of computing the summation in Eq. 3 with respect to all points, for each data point we update the features of vertices in a local neighborhood. The radial basis function $\phi$ decays exponentially, and so points beyond a local neighborhood have little influence (depending on choice of bandwidth $\gamma$). Restricting to a constant size local neighborhood improves computation from $O(NV)$ to $O(N)$.

To define the local neighborhood of data point $p$ in this work: any given point $p$ is contained within two bounding "triangles" of the discretization (ignoring boundary conditions and degenerate cases). These correspond to the vertices $S^{(i)} = \{u^{(i)}, v^{(i)}, w^{(i)}\}$ and $S^{(i+1)} = \{u^{(i+1)}, v^{(i+1)}, w^{(i+1)}\}$, where $i$ indexes radial level. However, using a single $\gamma$ value for the RBF results in scaling inconsistency: distances between vertices progressively shrink moving to inner spheres. Based on the assumption that RBF values should be invariant to scale, a different $\gamma$ and corresponding RBF is defined with respect to radial level. To define $\gamma_i$, we use the maximum pairwise distance $d_{max}^{(i)}$ between vertices in $\{S^{(i)}, S^{(i+1)}\}$. Specifically, we set $\gamma_i = \frac{-\log T}{d_{max}^{(i)}{}^2}$, where $T$ is a lower bound target RBF value. For example, $T = 1$ would correspond to $\gamma_i = 0$, or a RBF value of 1 at any distance. $T \in (0, 1]$ is a tuning parameter that enables toggling the overall sensitivity of the RBF to distances. Based on the approximation that $d_{max}^{(i)}$ is similar for any data point, $d_{max}^{(i)}$ is precomputed once.

## A.2  MODELNET40 TIME ANALYSIS

| Baseline | CSGNN | PointNet | RIConv | SPHNet | SRINet | SFCNN | PRIN |
|---|---|---|---|---|---|---|---|
| Training (hrs) | 5.1 | 3.3 | 1.9 | 2.4 | 4.1 | 4.8 | 6.1 |
| Inference (s) | 0.16 | 0.016 | 0.062 | 0.099 | 0.097 | 0.069 | 0.004 |

Table 6: Time comparisons of baselines for ModelNet40 dataset. Total training times are reported in hours. Inference time, in seconds, is for batch size of 32. CSGNN is our model.

We compare total training time and batch inference time of baselines from Sec. 5.1, for all instances where code was available. Results are reported in Table 6. All baselines were run on the NVIDIA Tesla P100 GPU. Total training time includes data loading and transformation time. Inference time is based on batch size of 32, and computed from averaging 32 different batches. The inference times reported do not include data loading or transformation time.

## A.3  CHOICE OF CENTER

| Center | Centroid | Component 1 | Component 2 | Component 3 |
|---|---|---|---|---|
| SO3/SO3 | 0.884 | 0.878 | 0.873 | 0.866 |

Table 7: Comparison of different choices of center. Centroid is the positional average of the point cloud. Components refer to the right singular vectors from singular value decomposition of the point cloud, decreasing order of singular value. The components serve as alternative centers. Performance is accuracy from training and testing with rotations.

A choice of center is needed to represent the 3D data using concentric spheres, and also is necessary to define rotations of the data. The ModelNet40 dataset from Qi et al. (2017) centers the data based

$([GC_{32} + BN + ReLU]_{L4} \times 2) + [RC_{32} + BN + ReLU]_{L4} + [RC_{64} + BN + ReLU]_{L4}$
$+ MaxPool$
$+ ([GC_{64} + BN + ReLU]_{L3} \times 2) + [RC_{64} + BN + ReLU]_{L3} + [RC_{128} + BN + ReLU]_{L3}$
$+ MaxPool$
$+ ([GC_{128} + BN + ReLU]_{L2} \times 2) + [RC_{128} + BN + ReLU]_{L2} + [RC_{256} + BN + ReLU]_{L2}$
$+ MaxPool$
$+ [GC_{256} + BN + ReLU]_{L1} + [RC_{512} + BN + ReLU]_{L1} + MaxPool$
$+ [GC_{512} + BN + ReLU]_{L0} + GP_{vertices} + [RC_{1024} + BN + ReLU]_{L0} + GP_{radial}$
$+ FC_{40} + Softmax$

Figure 5: Architecture used for Sec. 5.1 and 5.2 experiments. Layers proceed from left to right. $GC$ and $RC$ indicate graph and radial convolution layers, subscripted by output channel dimension. $BN$ is Batch Normalization, $FC$ is fully connected layer. $GP$ indicates global pooling with max operation. Subscript "$L$" indicates icosahedron level, corresponding to discretization level of the sphere.

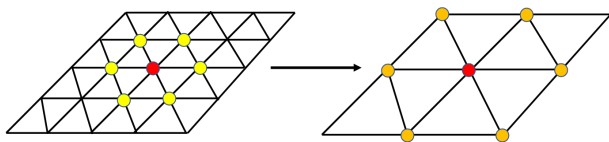

Figure 6: A vertex (red) and its neighbors (yellow/orange) are shown in local patch of the icosa-hedral spherical discretization. These are the basic units for intra-sphere convolution and pooling. Downsampling results in a coarser grid and vertex set, where a new neighborhood is defined (orange vertices)

.

on the centroid of a point cloud, while the SHREC17 dataset uses the centroid of mesh shapes. Our experiments in Sec. 5.1 and 5.2 follow this convention. In this section we also consider other choices of center besides centroid, and show that our representation is relatively robust to other centers of rotation, in application to the ModelNet40 dataset. We apply singular value decomposition to an input point cloud to obtain 3 principal component vectors. These 3 dimensional vectors each serve as a new reference point by which the point cloud is re-centered. Results from Table 7 show that centroid-based center produces performs best with our model. Performance of principal component-based center declines according to singular value. A possible reason for worse performance in the non-centroid center examples is that for the same number of points, a relatively smaller fraction of the concentric spherical discretizaton is utilized (following re-centering), effectively lowering the resolution of the input represenation. Overall the results suggest that CSGNN can be effective even when using non-centroid centers of representation and data rotation.

## A.4 ARCHITECTURE DETAILS

Fig. 6 provides visualization of the vertex pooling and downsampling operations. Fig. 5 provides more detail to the base architecture used in our experiments. Radial convolutions are implemented using 1D convolution layers with a kernel size of 3 in the ModelNet40 experiment, and either 1 or 3 in the SHREC17 experiment. Radial convolutions are followed by max pooling and downsampling to a lower level of spherical discretization, reducing the number of vertices. We also add residual connections between every graph convolution layer, whenever the number of input channels matches output channels.

## A.5 VISUALIZING CONCENTRIC SPHERICAL FEATURES

To better understand the impact of the multi-sphere aspect of the concentric spherical representation, we visualize the learned features from 3 different point cloud examples in Fig. 7. We use the 1st

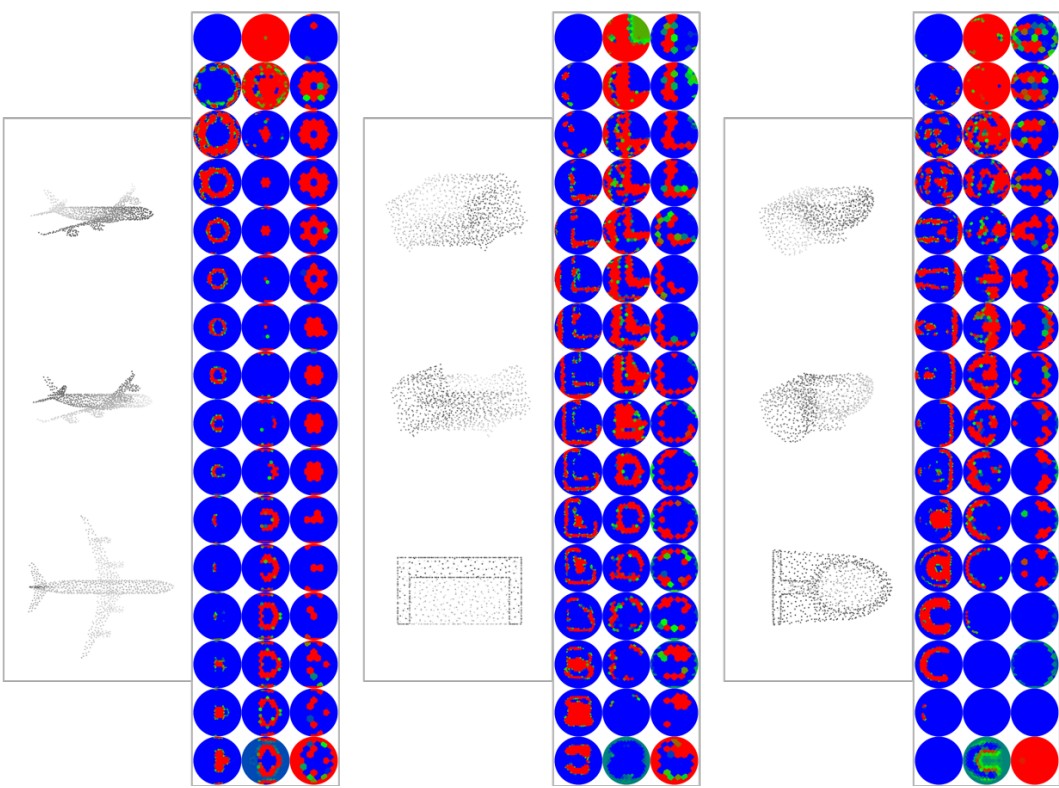

Figure 7: Visualization of point clouds from ModelNet40 and learned features. Example instances from left to right, shown in 3 different orientations: airplane, sofa, and toilet. In the table of spherical visualizations, each sphere corresponds to a single feature channel. Rows correspond to radial level (16 total), with bottom rows corresponding to outer spheres. Columns correspond to discretization level of the sphere, from level 4 to 3 to 2 (left to right). Colors are interpolated between blue and red, corresponding to low or high normalized feature values.

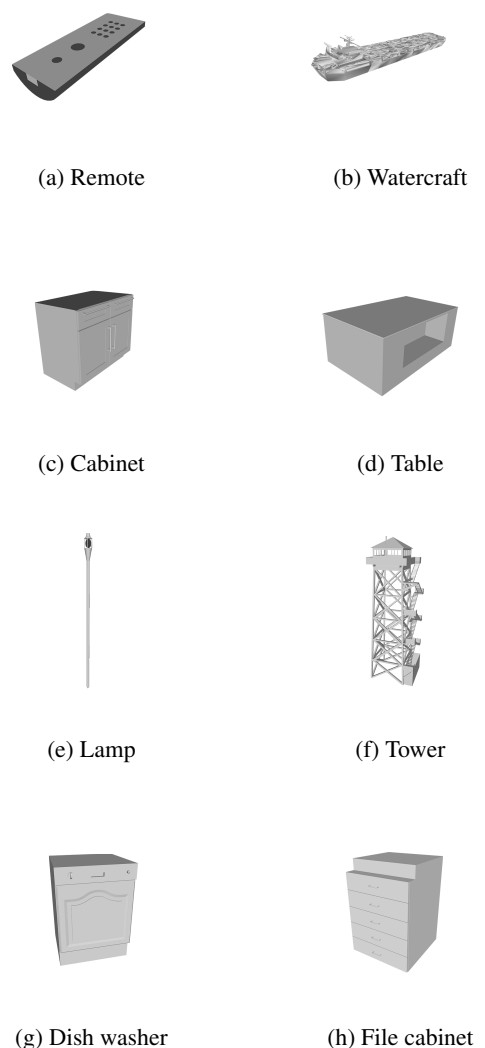

Figure 8: SHREC17 mis-predicted class pairs from single-sphere model where the multi-sphere ($R = 16$) model showed biggest relative improvement. Each image is a representative sample from the class. Note that *watercraft*, *table*, and *tower* all have more non-convex features that distinguish them from their mis-predicted counterparts. The concentric spherical model seems to better capture these differences.

feature channel in each visualization instance, and the features are obtained after radial convolution layers but before pooling and downsampling. The results visually demonstrate that different radial spheres within the same representation are capturing different features. At the same time, there is also a high degree of continuity between consecutive spheres in most cases, suggesting that there is information sharing between spheres resulting from radial convolutions.

A.6 SHREC17 VISUALIZATION

See Fig. 8 for SHREC17 visualization.

