# OpenReview forum: "Concentric Spherical GNN for 3D Representation Learning"
_ICLR.cc/2021/Conference — Reject_

### Official Review · AnonReviewer4 · 2020-10-27

**Rating:** 6
**Confidence:** 3

**Review:**

Summary:
This paper presents a novel multi-sphere icosahedral discretization for representation of 3D data. Given meshes or point cloud, the authors map them to multiple layers spheres and apply graph conv on the spheres to learn rotation-invariant features. In the final stage, all the layers are merged via a radial pool operator. The authors claim that such a structure could better preserve information of 3D objects and showcase on mesh and point cloud classification tasks.

Comments:
The authors extend the spherical CNN from a single layer to multiple layers and propose corresponding graph conv and pool operations for it. However, I have several concerns, as seen below:

Hierarchy v.s. partition: In my understanding, the authors use RBF to map each point to its nearest sphere vertex, as shown in Fig.1. Does it indicate the multilayer spherical CNN is not hierarchical but more about partition?  Also, RBF needs some explanation. I only see the abbreviation.

More explanation should be added in Sec.5.2, especially the way to do ray marching in multi-layer spherical CNN. It is a valid concern that the regular ray marching in Esteves 2018 would lose information for non-convex objects. But after reading the section, I am not exactly clear about how you preserve information for the multiple surfaces. Also, such new ray marching is more about hierarchy or partition?

In the experiments, I find the authors directly use the numbers in others' papers. This is not super fair since the experiments may be done in different settings. Why the number in Table 1(16 layers, SO3/SO3 is 0.884) and Table 3(16 layers, SO3/SO3 is 0.879?) are different?  Also, some time analysis is worth adding.

Conclusion: Overall, I think extending the spherical CNN from 1 layer to multiple layers is an interesting idea and one can foresee that such an extension would bring better performance, as shown in the paper. I think validating such an idea is a solid contribution and would bring insights to others. Thus I would like to accept this paper.

---

> ### Author Response · Authors · 2020-11-21
> **Response to Reviewer #4**
>
> We thank the reviewer for the feedback and questions. We attempt to address points raised in the review, below, organized topically:
>
> **Hierarchy vs. partition**: We assume that partition refers to the division of space into multiple spheres. The proposed architecture is still hierarchical, due to vertex-level pooling and coarsening of the spherical graph. It is also important to exchange information over multiple radial levels via radial convolutions, as the experiment in Table 4 shows. In other words, it suboptimal to all restrict convolutions to be within each partition.
>
> However, hierarchy does not seem necessary in the radial dimension itself.
> We conducted an additional ModelNet40 experiment where the radial dimension $R$ is gradually reduced (following convolutions) from 16 to 8 to 4 via max pooling. This led to worse accuracy of 86.6\% compared to best accuracy of 88.4\% from keeping the radial dimension fixed at $R=16$ through all convolution layers. We have not included this result in the paper for sake of space and concise description.
>
> In our view, the ray casting from Sec. 5.2 has to do with partition: ray casting is performed from each concentric sphere, and only hits in its own partition (the region of space between itself and the next inner sphere) are recorded.
>
> **Ray casting**: We re-wrote and added more detail to explain the ray-casting scheme from multiple spheres in Sec. 5.2.
> Our mapping does not guarantee that all multi-surface information is preserved.
> However, it provides more sampling of surface information by ray-casting from the same point rescaled to multiple spheres. In this way, the number of samples scaling linearly with the number of spheres introduced.
>
> **Performance difference in Table 1 (0.884) and Table 3 (0.879)**: this is due to difference in the function used in mapping points to vertices (in the conversion to concentric spherical features). The Table 1 version uses Gaussian radial basis function, whereas the Table 3 version uses a constant function. The difference in function is explained more in Table 4 (newly added).
>
> **Time analysis**:  an additional section has been added to Sec. A.2 of appendix, comparing total training time as well as batch inference time in the ModelNet40 experiment.
>
> "RBF" stands for radial basis function, which is discussed in Sec. 4. As an additional clarification, each point is not mapped via RBF to just the nearest spherical vertex, but a bounding neighborhood of 6 vertices (in the general case). This is neighborhood is re-illustrated in Fig. 1b and 1c for more clarity.

---

### Official Review · AnonReviewer2 · 2020-10-28
**In this paper, a multi-resolution convolutional architecture is proposed to learn from concentric feature maps. Different from single sphere representation, both graph convolutions and radial convolutions are employed to extract the intra-sphere and inter-sphere information.**

**Rating:** 6
**Confidence:** 4

**Review:**

In this paper, a multi-resolution convolutional architecture is proposed to learn from concentric feature maps. Different from single sphere representation, both graph convolutions and radial convolutions are employed to extract the intra-sphere and inter-sphere information. Benefit form the radial discretization, the proposed CSGNN achieves state-of-the-art results in testing on arbitrarily rotated data under ModelNet40 dataset. However, there are several drawbacks in the  draft, such as ambiguous figure and  insufficient ablation study.

## Summary

In CSGNN,  a  two-phase convolutional scheme for learning over a concentric spheres representation, by alternating between inter-sphere and intra-sphere convolutional blocks. Specifically, the graph convolutional network is applied to incorporate intra-sphere information, and 1D convolution to incorporate radial information. Different from previous works,  a multiple sphere representations are integrated in CSGNN, which is robust to rotation operations. In general, the proposed method is innovative on the basis of spherical discretization, but still has some limitations.

## Strengths

#### 1. A multi-sphere icosahedral discretization for representation of 3D data is proposed, which can enhance  representation ability and keep more details over single-sphere representations.

#### 2. Both graph convolutional networks and 1D radial convolutions are employed to capture the intra-sphere and inter-sphere information of 3D objects.

#### 3. The proposed CSGNN achieves state-of-the-art results in testing on arbitrarily rotated data.

## Weaknesses

#### 1. The complexity analysis is insufficient. In the draft, the author only provide the rough overall complexity. A better way is to show the comparison between the proposed method and some other methods, including the number of model parameter and network forwarding time.

#### 2. In the converting of point cloud to concentric spherical signal,  the Gaussian radial basis function is adopted to   summarize the contribution of points. Is there any other function that can accomplish this job? The reviewer would like to the discussion about this.

#### 3. The Figure 2 is a little ambiguous, where some symbols are not explained clearly. And the reviewer is curious about that  whether there is information redundancy and interference in the multi-sphere icosahedral discretization process.

#### 4. There are some typos in the draft. The first is the wrong use of "intra-sphere" and "inter-sphere". The second is the use of two consecutive "stacking" in the Spherical Discretization subsection. Please check the full text carefully.

#### 5. The center choice of the concentric spheres should be discussed both theoretically and experimentally. In the opinion, the center of spheres play a important role in the representation capturing of 3D point clouds in a sphere convolution manner.

---

> ### Author Response · Authors · 2020-11-21
> **Response to Reviewer #2**
>
> We thank the reviewer for the feedback and suggestions. Below are our responses to different points organized in order of items listed in "Weaknesses" section.
>
> 1.  **Complexity analysis, comparing parameters and network forwarding time**:
> We have added the number model parameters for all baselines in both the ModelNet40 and SHREC17 experiments, to Table 1 and 2 respectively.
> We have also added a comparison of batch inference time, as well as total training time, for baselines from ModelNet40 experiment--see Sec. A.2 of Appendix.
> We have also reorganized complexity analysis into separate paragraph in Sec. 5, and added some more detailed explanation.
>
> 2. **Alternative functions to summarize contribution of points**: We have added a new section, Sec. 5.4, that explores alternatives to the Gaussian radial basis function.
> While the Gaussian radial basis function is still best performing, the alternatives perform comparably.
>
> 3. **Information redundancy and interference in discretization process**: We assume that by discretization process, the reviewer is referring to mapping of 3D data to the multi-spherical icosahedral discretization.  We separate the discussion for two cases: point cloud and mesh object inputs.
>
>     1. For point cloud data, there is some redundancy in initial features of the multi-spherical representation.
> To review, for a given vertex, contributing points in its local neighborhood are summarized into
> a feature vector.
> This local neighborhood spans a partition of space bounded by the sphere the vertex belongs to, as well as the sphere below (see Fig. 1b for visualization).
> The current implementation concatenates neighborhood information from *two* consecutive partitions per vertex, and so there is a 1/2 factor overlap between initial features of a vertex and its counterpart in the sphere below.
> There may be some interference resulting from summing contributions of multiple points per vertex, as there is a smoothing effect that could become too large with the number of neighborhood points.
> However, we believe this is unlikely to be an issue with sufficient resolution of the proposed discretization.
>
>     1. For mesh data: we don't expect data redundancy or interference here, as each vertex generates a unique ray, such that there is a one-to-one mapping between intersection point with the mesh (if it exists) and each vertex.
> This is achieved by filtering intersection points that do not belong to the same partition as the vertex that is the source of the ray in our implementation.
>
> 4. We  thank the reviewer for pointing out typos and points of ambiguity in the paper, and will work on fixing these in the draft.
>
> 5. **Choice of center**: The datasets used in this work (ModelNet40, SHREC17) were preprocessed to use the centroid of input 3D data as choice of center, a convention that other baselines we compared with follow.
> We have added experiments in Sec. A.3 of Appendix to compare how our model performs if different (non-centroid) choices of center are used.

---

### Official Review · AnonReviewer3 · 2020-10-29
**Review of "Concentric Spherical GNN for 3D Representation Learning"**

**Rating:** 5
**Confidence:** 4

**Review:**

This paper addresses an important problem in 3D representation learning, which is how to design a robust convolution neural network for arbitrarily oriented inputs. They propose to use multi-sphere icosahedral discretization to representation 3D data at first. And then, several alternative convolution operations are used to further process features. The novel convolution operation is consisting of intra-sphere and inter-sphere convolution. They also design a mapping function to convert point cloud data to mesh data. The experimental results indicate the model can work well.

Strengths:
1.	A novel method to discretize space, which can be used for represent 3D data.
2.	The convolution combines intra-sphere information with inter-sphere information, it can keep rotation invariant.
3.	New mapping is proposed to convert point cloud data to mesh data, it can make the model more general to deal with 3D data.
4.	Results show the model is effective and achieves SOTA.

Weakness:
1.	Math symbols need to be unified.
2.	Some repeated words. E.g. “stacking” in “Radial Discretization”, Page 3, “point cloud” in “POINT CLOUD TO CONCENTRIC SPHERICAL SIGNAL”, Page 4.
3.	Some math symbols have no explanation, which may make readers cannot get your idea. E.g. d_u、d_v in Eq(1)。
4.	I don’t know what is Z_g in Eq(1), and the shape of this parameter. The oversimplified description of Z_g makes me can’t understand why Eq(1) can keep rotation invariant.
5.	What is the shape of W_{k+K/2} in Eq(2), is it a matrix ? Please add more description of your math symbels.
6.	How to determine the neighbor range of point x in Eq(3) ? Are you use KNN or ball query methods ? Do you have some contrast experiments?
7.	Only use the norm-2 to calculate Eq(3) may loss some direction information, how do you think of this question?

Although the paper seems to propose an effective convolutional operation for point cloud representation, it is unclear how to maintain its rotation invariance. Possibly this is because of inconsisent and unclear symbols. Therefore, if the authors could show this factor satisfactory in the rebuttal and address the above concerns, I would like to move to a positive rating.

---

> ### Author Response · Authors · 2020-11-22
> **Response to Reviewer #3**
>
> We thank the reviewer for the feedback and questions. We attempt to address the reviewer's concerns below, organized topically:
>
> **Confusion over math symbols and parameters**: We apologize for the inconsistent use of notation, and lack of description in some cases.
> We also noticed errors in the convolution equations that may have impacted reviewer's understanding of them.
> We have re-worked the symbols and their explanations for Eq. 1 and 2 (intra-sphere and inter-sphere convolutions).
> $Z$ and $W_{k+K/2}$ are both matrices of parameters. The dimensions of $Z$ and $W$ have been explicitly defined in Sec. 3.
>
> **Maintaining rotational invariance**:
> We separate the discussion of rotational invariance for intra-sphere and inter-sphere convolutions.
>
> For intra-sphere convolution, we have added reference to a recent paper [1], here and in the paper, to add to discussion of rotational invariance of graph convolutions on the icosahedral sphere.
> [1] showed that graph convolutions with respect to the icosahedral sphere are equivariant to the icosahedral symmetry group, as graph isometric transformations.
> Equivariance is maintained through pooling operations, and invariance can be resolved through global pooling.
> The icosahedral symmetry group is a subset of SO(3), and so strict  rotational invariance is limited by effects of discretization.
> While our choice of graph convolution in differs from the formulation in [1], we believe that our proposed approach is flexible to the specific choice of graph convolution.
>
> We proposed radial convolution for inter-sphere convolution, which uses 1D convolution by treating the radial dimension as the sequence dimension.
> The 1D convolution operates only on channel information of the same vertex (recall the multi-radius discretization is formed by stacking the same spherical discretization), which is rotationally invariant (up to effects of discretization).
>
> [1] Yang et. al. "Rotation equivariant graph convolutional network for spherical image classification". CVPR'20.
>
> **Picking neighborhood range**: We have added a new section, Sec. 5.4, that explores neighborhood selection.
> Our current method takes a point-centered approach, i.e. each point influences its immediate neighborhood of vertices.
> This neighborhood is visualized in Fig. 1b (newly added), and consists of 6 vertices. The exception is for points within the innermost sphere, where there are only 3 vertices in the neighborhood.
> The influences from all points over vertex neighborhoods are summed.
>
> The point-centered approach, which uses "bounding vertices" neighborhood, is not equivalent to $k$-NN nor ball query. As an alternative, we also added a vertex-centered $k$-nearest neighbors approach for comparison, where neighbors are points.
> Results from Table show that that the point-centered bounding vertices approach is performing better.
>
> **Loss of directional information using 2-norm**: The reviewer makes a valid point that original directional information of points is lost using l2-norm when assigning to vertex values using radial basis functions.
> The choice of mapping points to vertices in this way is a trade-off with rotational invariance concern, as directional information is not invariant to rotation.
> Despite loss of directional information using l2-norm, each point influences a neighborhood of vertices in 3D space.
> If we consider the case of a single point influencing its neighborhood (see example in Fig. 1b,c), changes in the point's position results in a change in the distribution of values on those vertices, and so positional information is recovered to some degree.
> However, this likely would not be sufficient with multiple points mapping to same vertices, due to effects of summing.

---

### Official Review · AnonReviewer1 · 2020-10-29
**well presented and reasonable results; lack of sufficient details and not demo video for 3D visual results**

**Rating:** 5
**Confidence:** 3

**Review:**

This paper proposes a Spherical GNN approach for effective representation of 3D data (point clouds and surface meshes).
The main ideas are interesting and reasonably well presented. The results are convincing, being comparable to or better than SOTAs on benchmarks.
However, I am not an expert in this area: It seems to be an interesting new idea proposed in this work; meanwhile, I feel a lack of sufficient technical details for reproduction.
It would be helpful if the authors could explain some of the key ingredients, such as the radial convolutions, inter/intro sphere convolutions in greater details. The proposed NN architecture, as illustrated in Fig. 3, still lacks in many details, and possibly is not sufficient for a faithful reimplementation/reproduction by other researchers.
The authors should make their code and processed data and results publicly available.
The empirical evaluation is also somewhat lacking in its breadth and depth. The shapenet, for example, also contains lots of 3D shapes of various types of objects, and could be a good dataset for evaluaiton. In terms of applications, it would be helpful to showcase a few application studies and to demonstrate the advantages of adopting the proposed scheme.
The authors should also present more visual results of their 3D shape representation in a supplementary video.

---

> ### Author Response · Authors · 2020-11-25
> **Response to Reviewer #1**
>
> We thank the reviewer for the feedback and suggestions. We address the reviewer's feedback to the best of our understanding and ability, below:
>
> **Lack of sufficient detail for reproduction**: To address this concern, we have re-worked the paper in several directions. (1) We have included a more detailed per-layer breakdown of the model architecture in Sec. A.4 and Fig. 5 of the Appendix, which should be more useful for re-implementation.
> (2) We have re-written the "Intra-sphere Convolutions" and "Inter-sphere Convolutions" subsections of Sec. 3 to explain the convolutions more clearly.
> This includes re-working the mathematical notation and fixing inconsistencies that may have caused confusion.
> (3) We have added an additional "Vertex Pooling" subsection in Sec. 3, along with an associated visualization in Fig. 6 of the Appendix, as this aspect was used but not explained in detail previously.
>
> **Empirical evaluation breadth and depth**:
> To address breadth of evaluation of the proposed method, we conducted two classification  experiments for two different types of 3D data: raw point clouds, and mesh objects files.
> We also proposed very different data transformation strategies in order to map these respective input formats to concentric spherical feature maps.
>
> We clarify that our experiment Sec. 5.2 is using the ShapeNet Core55 dataset, which we referred to as "SHREC17" in the paper in reference to the competition [1] where this dataset was used.
> This competition subset of ShapeNet has advantages of standardized preprocessing and predefined rotated versions of objects, as well as predefined train/validation/test splits. We have also added additional experiments on picking neighborhood range, choice of center, and alternative mapping functions.
>
> **Demo video for visual results of 3D shape representation**: We have created a video to present visual results of our representation in two different ways.
> The first is explaining step-by-step our point cloud to concentric spherical discretization mapping, as well as intra-sphere and inter-sphere convolutions, using visual example.
> We believe this can be generally helpful for conceptually visualizing the proposed representation and convolutions.
> The second way is showing visual examples of learned representations using our model, by plotting feature activations.
> These result from real point cloud examples from ModelNet40 dataset, and highlight the concentric spheres contribution.
>
> **Making code public**: We are in the process of making the code public, with approval for code release in progress.
> These visualizations are also included in Sec. A.5 of Appendix.
> Please see supplementary material for the video.
>
> [1] Savva et. al. "Large-Scale 3D Shape Retrieval from ShapeNet Core55: SHREC’17 Track". 3DOR'17.

---

### Author Response · Authors · 2020-11-25
**Summary of changes**

We thank the reviewers for their feedback, suggestions, and questions for helping to improve this work.
We have highlighted more substantial revisions to the paper in red.
We also summarize below the main updates and additions we have made through this process:

1. We have re-written large parts of "intra-sphere convolutions" and "inter-sphere convolutions" subsections of Sec. 3 in order to explain the proposed convolutions in more detail. This includes re-working and unifying math symbols to remove confusion as much as possible.

2. Added more detail for reproducibility: we added "vertex pooling" subsection to Sec. 3 to explain pooling operation used in this work in more detail, along with a visualization in Fig. 6 of Appendix.
We also added sections A.4 and Fig. 5 in Appendix to provide more detail on the proposed architecture.

3. We added additional experiments and their discussion in the following places: (1) Sec. 5.4 and Table 5 for comparing choice of neighborhood, as well as alternatives to Gaussian radial basis function. (2) Sec. A.3 of Appendix for comparing alternative choices of center.

4. We added comparison of runtimes in Sec. A.2. of Appendix, and number of parameters to Tables 1 and 2.

5. Added visualization of our learned representation by plotting features obtained from convolutions, in Fig. 7 and Sec. A.5 of Appendix.

6. Updated Fig. 1 to explain visual example of mapping of point cloud to concentric spherical discretization more clearly.
We also added Fig. 2 to provide better visualization of intra-sphere and inter-sphere convolution concepts.

7.  Submitted demo video as supplementary material, presenting on 3 different aspects: (1) point cloud to concentric spherical discretization mapping (2) intra-sphere and inter-sphere convolutions (3) visualization of learned features from example point clouds.

---

### Decision · Program_Chairs · 2021-01-07
**Final Decision**

**Decision:**

Reject

**Comment:**

The paper proposes  to effectively learn representation of 3D data (point clouds/meshes) using a spherical GNN architecture over concentric spherical maps. A method for converting point clouds to concentric spherical images is also proposed. Evaluation is done via 3D classification tasks on rotated data.

Strengths:
- Interesting novel method for learning 3D representations
- Technically sound
- Performs similarly to spherical CNNs and other STOA on the Modelnet40 dataset

Weaknesses:
- Presentation of the work needs to be further improved such that it is easier for others to reproduce
- More in-depth experiments are needed to justify how much Spherical GNN improves over other STOA, particular given how classification accuracy is very similar to STOA.